# Robust thermoelastic microactuator based on an organic molecular crystal

Yulong Duan[1], Sergey Semin[1], Paul Tinnemans [1], Herma Cuppen [1], Jialiang Xu[2]* & Theo Rasing[1]*

Mechanically responsive molecular crystals that reversibly change shape triggered by external stimuli are invaluable for the design of actuators for soft robotics, artificial muscles and microfluidic devices. However, their strong deformations usually lead to their destruction. We report a fluorenone derivative (4-DBpFO) showing a strong shear deformation upon heating due to a structural phase transition which is reproducible after more than hundred heating/cooling cycles. Molecular dynamic simulations show that the transition occurs through a nucleation-and-growth mechanism, triggered by thermally induced rotations of the phenyl rings, leading to a rearrangement of the molecular configuration. The applicability as actuator is demonstrated by displacing a micron-sized glass bead over a large distance, delivering a kinetic energy of more than 65 pJ, corresponding to a work density of 270 J kg$^{-1}$. This material can serve as a prototype structure to direct the development of new types of robust molecular actuators.

[1] Radboud University, Institute for Molecules and Materials (IMM), Heyendaalseweg 135, 6525 AJ Nijmegen, the Netherlands. [2] School of Materials Science and Engineering, National Institute for Advanced Materials, Nankai University, Tongyan Road 38, 300350 Tianjin, P.R. China. *email: jialiang.xu@nankai.edu.cn; th.rasing@science.ru.nl

Thermosalient crystals are characterized by thermally induced solid-state phase transitions accompanied by a sudden anisotropic lattice expansion that induces a jumping, bending, deformation or rotation of the crystals upon heating[1–5]. They belong to a large class of dynamic crystals that respond to external stimuli, such as heat, light, or pressure, with various switching effects of their electronic[6,7], magnetic[8,9], optical[10], and mechanical[11–13] properties. These phase transitions are generally considered to proceed through cooperative molecular motions inside the crystal[3,5], but their understanding is still in its infancy. It is not evident how the classifications and phase transition mechanisms in inorganic crystals or crystals with much less molecular complexity[14,15] translate to molecular crystals, with weaker interactions and more steric hindrance. The large anisotropic changes in crystal shape and size are difficult to accommodate for the often brittle organic crystals and the mechanical effects are usually accompanied by crystal cracking, splitting or even explosion[1,2]. Therefore, new model crystal structures need to be developed that combine large thermomechanical effects with robustness of the crystal.

The known thermosalient materials were classified by Sahoo et al.[2] into three classes: two classes of layered materials, one containing flat, rigid molecules with weak interactions between them, the other containing molecules with strong interconnecting hydrogen bonds within the layer and both with weak interactions between the layers; the third class contains bulky flexible molecules with hindered rotations. Here we present a thermosalient material based on 2,7-di([1,1′-biphenyl]-4-yl)-fluorenone molecules (4-DBpFO)[16], which does not fall in any of these classes. The transition appears to be triggered by small conformational changes in the molecules, which propagate through the crystal and result in a large crystal shape change[2,3]. Practical application of 4-DBpFO is demonstrated by its ability to displace glass plates and a micron-sized glass bead over a considerable distance upon a slight temperature increase.

## Results

**Structural phase transition.** 4-DBpFO is constituted of five flat segments: a fluorenone center with two phenyl rings at each side

linked by single bonds (Fig. 1). The small rigid planes can rotate around the single bonds with a large degree of freedom (dihedrals θ and φ) when the molecules are heated in solution, similar to the molecular motors designed by Feringa's lab[17,18]. We have earlier reported that this compound crystallizes in a non-centrosymmetric crystal structure at room temperature with a $Cmc2_1$ space group (referred to as the α′-phase), useful for optical second harmonic generation[16]. Here, we show another room temperature phase of 4-DBpFO described by a centrosymmetric $Pnma$ space group (referred to as the α-phase), crystallized from a mixed solvent of chloroform and heptane (see Table 1 for cell parameters). The morphology of the as-grown crystals is a parallelogram-like thin platelet (Fig. 1a) with {010} basal planes and {101} or {10-1} side faces (Fig. 1b). The crystal structure shows an antiparallel molecular organization along the crystallographic b-axis (Fig. 1c), resulting in a layered structure parallel to the (010) plane (Fig. 1b, c). The corner angles of the parallelogram are 91° and 89°, in accordance with those calculated from the cell parameters at room temperature.

When a thin, almost square, parallelogram α-phase single crystal is heated to around 189 °C, it transforms to a new phase (referred to as the β-phase) with a more oblique morphology (Fig. 1a, Supplementary Movie 1). Depending on crystal quality the transition temperature may vary. This dramatic shape change can be understood as a shear deformation in the (010) planes, parallel to the < 101 > or < 10-1 > directions. To exclude thermal expansion effects on the cell parameters, the unit cell parameters of the α-phase at 170 °C (right below the phase transition temperature) and the β-phase at 178 °C (right above the phase transition temperature) were determined by single crystal X-Ray Diffraction (SXRD) (Table 1). To facilitate comparison with the α-phase, the high temperature β-phase is described in a non-standard setting. During the α- to β-phase transition, the b-axis stays almost the same; the a-axis shrinks and the c-axis expands, while the corner angles of the crystal change drastically (Table 1), leading to the observed crystal shear deformation in the (010) plane[19]. This unit cell change corresponds very well with the shape change observed in optical microscopy[20] (Supplementary Fig. 1). During this in-plane anisotropic expansion (Fig. 1b), the crystals still keep the layered structure along their normal direction (Fig. 1c). This shear

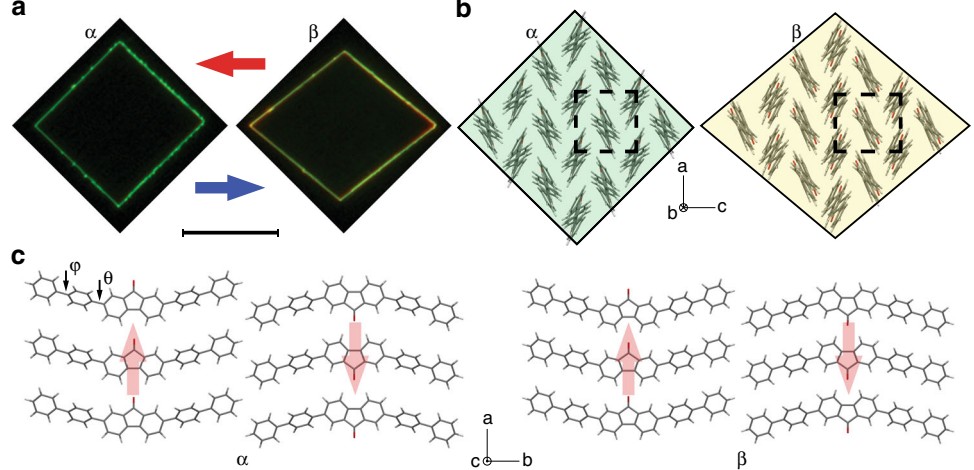

**Fig. 1** Crystal structure and morphology of the α-phase and β-phase 4-DBpFO. **a** Fluorescence images of a typical macroscopic single crystal of the α-phase (at room temperature) and β-phase (at 178 °C), under UV light excitation. Scale bar is 50 μm. **b** Molecular packing viewed along the b-direction of the two phases. The dashed frame shows the in-plane crystal lattice. **c** Molecular packing viewed along the c-direction. The molecular arrangement shows an overlapping layered structure along the b-axis with anti-parallel dipoles. The light red arrows show the direction of the molecular dipoles. The structure shows that there is ample space between molecules, which is beneficial for the rotation of the flat segments in the molecules during the phase transition. Notice that the angles between the axes are all 90° in the α-phase while the α-angle is 93° in the β-phase. φ is the dihedral angle between the inner and outer phenyl rings and θ is the dihedral angle between the inner phenyl rings and the rigid center

deformation is very pronounced compared to that found in photosalient crystals. For example, Kobatake et al. reported a diarylethene crystal which can undergo a similar shear deformation by light irradiation, but the corner angles only changed from 88º and 92º to 82º and 98º.

The changes observed in our crystal are so large and abrupt that, when the crystal was placed sideways on one of its {101} or {10-1} facets, it jumped far away upon heating (Supplementary Fig. 2). When lying on its basal plane a distinct and coherent phase boundary between the α- and β- phases was clearly observed during the phase transition, starting from one of the {101} or {10-1} facets, as shown in Fig. 2a. Remarkably, this phase boundary propagated through the entire crystal upon continuous heating, being always parallel to one of the crystal sides and staying macroscopically undistorted during migration through the crystal. There is no strong interaction between the molecules within the layers and this allows the phase boundary to be perpendicular to the layer plane, in contrast with two of the classes by Sahoo et al.[2]. The third class includes transitions with hindered rotation of bulky groups. Although torsional rotation plays a role in the present transition, it is not hindered since the dihedrals do not change sign. This resembles the characteristics of martensitic transformations in metals[21], which also proceeds by

**Table 1 Cell parameters of the 4-DBpFO crystal at different temperatures**

| Phase | | α | | β |
|---|---|---|---|---|
| Crystal system | | Orthorhombic | | Monoclinic |
| Space group | | *Pnma* | | *P2₁/n* |
| Temperature (°C) | | 25 | 170 | 178 |
| Cell lengths (Å) | a | 6.8552(6) | 6.90 | 6.256(5) |
| | b | 51.777(5) | 52.51 | 52.72(4) |
| | c | 6.9557(6) | 7.27 | 7.773(6) |
| Cell volume (Å³) | | 2468.87 | 2632 | 2560.26 |
| Cell angles (°) | α | 90 | 90 | 92.95(2) |
| | β | 90 | 90 | 90 |
| | γ | 90 | 90 | 90 |
| Dihedral angle (θ) | | 38 | | 22 |
| Dihedral angle (φ) | | −24 | | −16 |
| Corner angles of the crystals | | 91°/89° | 88°/92° | 77°/103° |

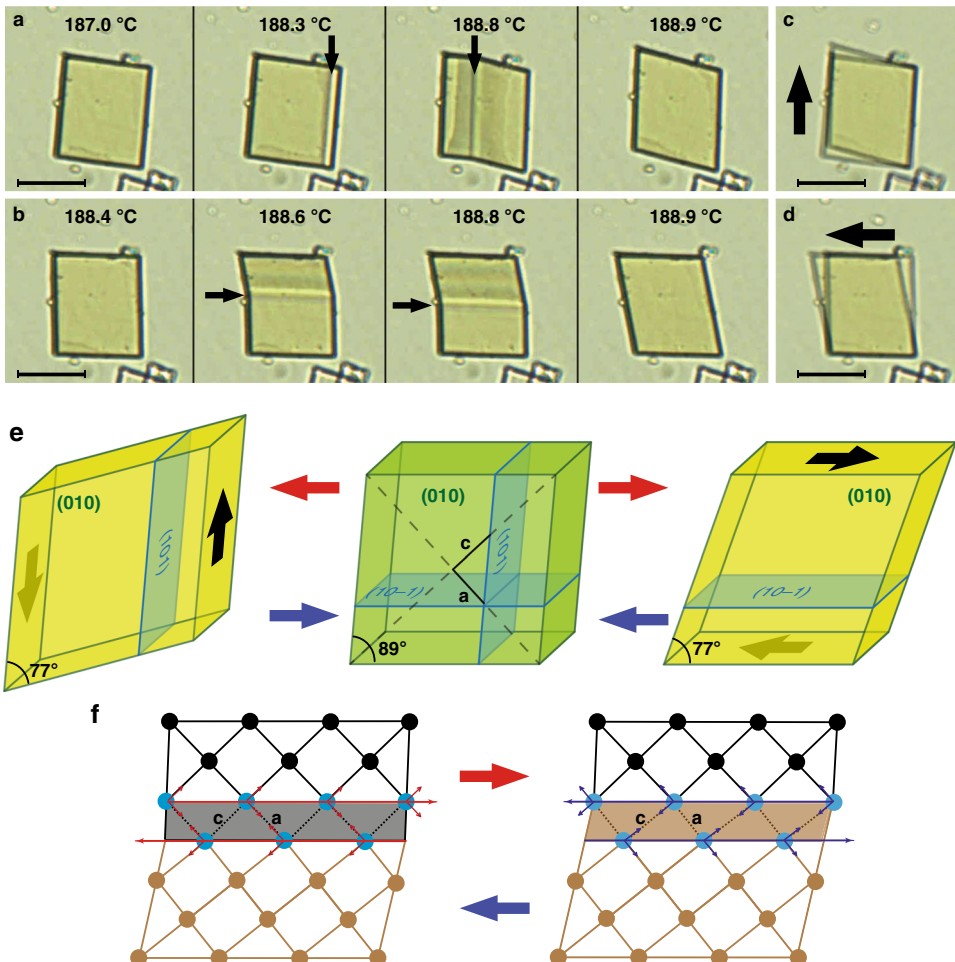

**Fig. 2** Two directions of shape deformation during the phase transition. **a, b** Optical microscope images show that the temperature dependent shape deformation from the α- to β-phase can proceed in two independent directions. The position of the phase boundary is indicated with black arrows. **c, d** Overlapped microscope images of the same crystal before and after the phase transition, showing that the crystal orientations relate to the direction of the phase boundaries. The arrows indicate the corresponding shear directions. Scale bar is 20 μm. **e** Schematic presentations of the corresponding crystallographic planes and the shear directions (black arrows). The blue planes indicate the possible directions of the phase boundaries, which are parallel to the **(101)** or **(10-1)** plane. **f** Schematic presentations of the origin of the shear force and its relation to the shrinking/contraction (a-axis) and contraction/shrinking (c-axis) in the **(010)** plane during heating (short red arrows)/cooling (short blue arrows). The black and the gray grids represent the lattices of the α-phase and the β-phase, respectively. The long red and blue arrows show the shear forces that result in the migration of the phase boundary during heating and cooling, respectively

the movement of an invariant phase boundary, called the habit plane. Generally, the structural transformation in thermosalient organic crystals will rapidly transform the whole crystal once initiated, which makes it hard to control. However, we found that the migration of the phase boundary of 4-DBpFO could be pinned at a certain position by keeping the crystal at a constant temperature, and its propagation direction could be even reversed upon cooling (Supplementary Fig. 3). This is presumably due to the presence of defects or dislocations[22]. As the **(101)** and **(10−1)** faces are alike due to pseudosymmetry (symmetry operation −z, −y, −x), the phase boundary can in principle appear in both directions of the same crystal, which was indeed observed in some crystals (Fig. 2a, b). The phase transition proceeding via the two different paths resulted in the same final crystal shape, in a mirrored orientation (Fig. 2c, d).

**Shear deformation and durability**. The observed shear deformation can be understood from the relationship between the crystallographic orientations of the two phases during the phase transition (Fig. 2e, f). The anisotropic lattice expansion in the **(010)** plane generates a shear force parallel to the unit cell diagonals, resulting in a huge macroscopic shear deformation parallel to the crystal sides of more than 18% (Fig. 2f and Supplementary Fig. 4 for method of calculation). This large strain is two times larger than that of thermally activated shape memory alloys (~8.5%)[23,24]. Although the cell lengths of the two axes in the **(010)** plane change prominently (~10%) during the phase transition, the diagonal lengths of the in-plane unit cell do not change and stay at

about 10.0 Å. Accordingly, the macroscopic lengths of the parallelogram-shaped crystal sides do not change. For martensitic phase transitions, the lattice parameters usually change prominently in all three directions, so an additional lattice invariant shear in the new phase is needed to maintain the invariant plane of the phase boundary[25,26]. This lattice invariant shear results in the formation of substructures in metals or alloys[27,28] which can be plastically deformed by twinning or slipping[29] (Supplementary Fig. 5). For brittle organic crystals, this lattice invariant shear often results in breakage, splitting or even explosion. For 4-DBpFO, the phase boundary is parallel to the **(101)** or **(−101)** plane during the migration (Fig. 2). As the lengths of the unit cell diagonal in the **(010)** plane and the normal $b$-axis stay the same, the phase boundary is an invariant plane and thus the phase transition does not need any extra lattice invariant shear[30]. This explains the full reversibility of the phase transition: the high temperature phase crystal returned completely to its original shape when cooled below 178 °C. This reversibility is also apparent from the differential scanning calorimetry (DSC) measurements (Fig. 3a): the change in enthalpy estimated from the DSC curve is 1.02–2.10 kJ mol$^{-1}$ for both cooling and heating. More importantly, the crystal showed full reversibility of the shape deformation after more than 100 cycles without any visible damage (Fig. 3b), demonstrating excellent fatigue durability of 4-DBpFO as a thermosalient material. Another reason for the good reversibility may be related to the fact that the in-plane anisotropic lattice expansion results from the in-plane rotation of the benzene ring planes in the molecules, while the rigid fluorenone center does not rotate. As shown in Fig. 3c, the dihedral angles θ and φ change, but their signs stay the

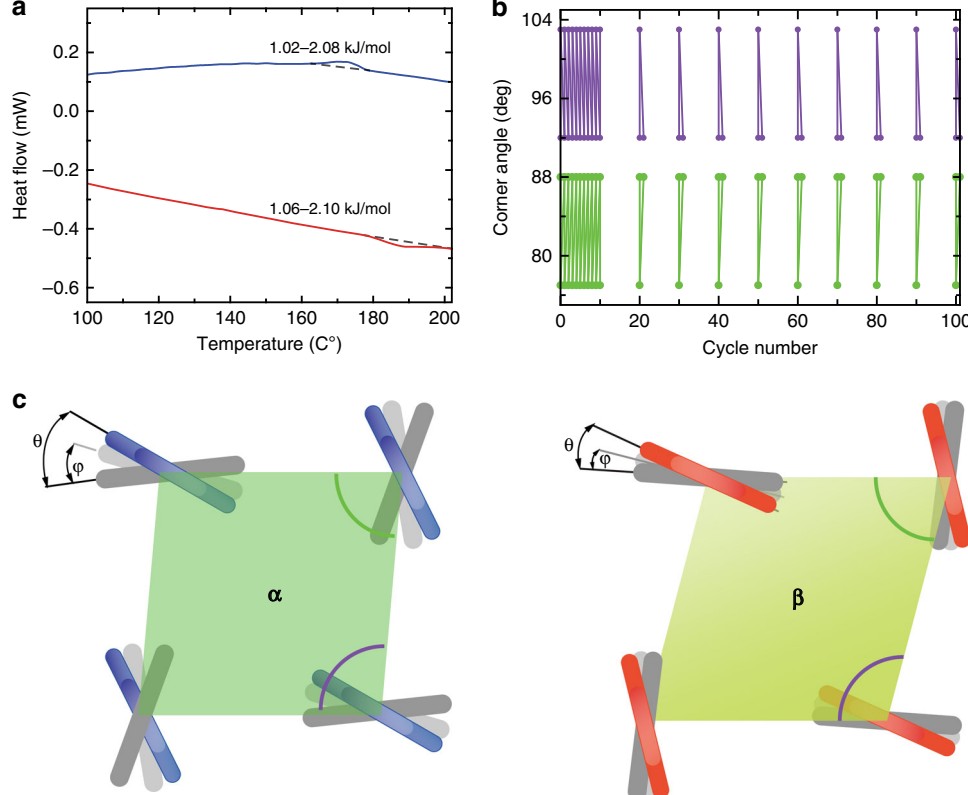

**Fig. 3** Reversibility of the shape deformation. **a** The DSC curve during the α- to β-phase transformation. The dashed lines are the baseline for the integration of the exothermic and endothermic peak. The same integral value indicates a completely reversible phase transition. Source data are provided as a Source Data file. **b** The reversibility of the macroscopic deformation, as measured by the reversible changes of the corner angles of the crystal, from 92° to 103° and from 88° to 77°, respectively, for over 100 heating/cooling cycles. **c** Schematic diagram of the molecular conformational changes during the phase transition. Rotation of the phenyl rings (gray segments) with respect to the rigid center part of the molecule (red and blue segments) leads to different corner angles

same after the phase transition (please refer to Table 1 as well). The molecular configurations become more planar after the phase transition, which may be the reason that the cell volume of the $\beta$-phase is 2.7% smaller than the $\alpha$-phase. In fact, crystal splitting parallel to the (010) plane was observed in a few crystals (two crystals out of 50+ crystals studied) which may be because of the weak intermolecular force between the layers along the normal direction; upon cooling, their shape still completely recovered, but not the damage due to the splitting (Supplementary Fig. 6).

**Molecular dynamics simulations of phase transition**. To gain more insight into the microscopic origin of the strong shape deformation of the 4-DBpFO crystal, classical molecular dynamics simulations were performed using LAMMPS. The simulation cell consists of four 4-DBpFO layers in the $y$-direction, each with four dihedral angles per molecule. The system was first equilibrated in the $\alpha$-phase at 27 °C and then kept at this temperature for 60 ps. As can be seen in Fig. 4b, the cell lengths of the $\alpha$-phase were well retained, as was its orthorhombic character, the molecular conformation (Fig. 4c, d) and the crystal packing, giving confidence to the applied force field. The system was subsequently heated during 100 ps to 177 °C (vertical dashed lines and color gradient background). The crystal showed roughly isotropic expansion during this heating, as is visible in the graph. After heating to 177 °C, a drastic change in the corner angles (Fig. 4a) and lattice parameters (Fig. 4b) can be observed. This is accompanied by changes in the dihedral angles of the molecules, as visualized in the bottom two panels. At elevated temperatures, the dihedral fluctuations of the outer phenyl rings increase and trigger a conformational change (phase transition).

Judging by the packing, the cell parameters and the molecular conformation, the system has transformed to the $\beta$-phase, in agreement with the experimental findings. We would like to stress here, that this transformation appeared in our simulations spontaneously: we did not steer the system by adding a bias, fixing the lattice parameters to the new form or changing the molecular conformation. Only, the volume was allowed to change and the molecules were treated flexibly while heating the system. A movie as well as snapshots of the onset of the transformation (Supplementary Movie 2) shows that within the simulation cell, a nucleation point of a few molecules can be identified after which the transformation spreads through the rest of the crystal in all three directions. In 15 ps the whole cell is transformed to the new structure. The general consensus in the literature is that thermosalient materials transform through a fast, cooperative mechanism[2,31]. Although the length scale of this cooperative movement is typically not specified, it is generally assumed to be on a larger scale than the $54 \times 108 \times 58$ Å scale of our simulation cell. The nucleation-and-growth mechanism that follows from our molecular dynamics simulations has the following consequences, in agreement with our experimental observations: (i) there is a well-defined transition temperature with little hysteresis and (ii) the transition is more gentle, making it less destructive and allowing it to be cycled many times without destroying the crystal.

**Temperature controlled movement of load**. To demonstrate the applicability of the 4-DBpFO thermosalient crystal, we used it to move a micron-sized glass bead by fixing the crystal on the substrate on one side and following the position of the bead with a CCD camera with a frame rate of 30 fps (see Fig. 5 and Supplementary Movie 3). Heating or cooling the crystal ($200 \times 200 \times 50\ \mu m$) through the transition both gave rise to a strong "kick" to the bead which subsequently was displaced several centimeters. From the measured average velocity (29 mm s$^{-1}$) and mass

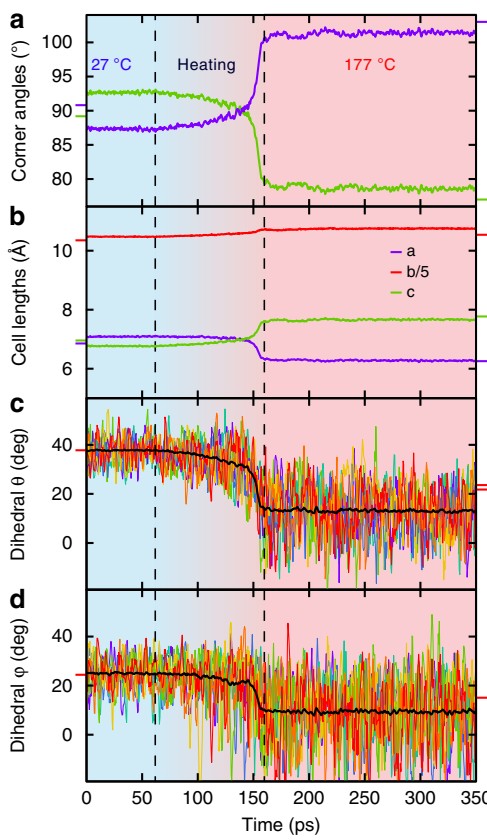

**Fig. 4** Molecular dynamics simulations. Molecular dynamic simulations show that the molecular configuration rearrangement by rotations of the dihedral angles (**c**, **d**) is accompanied by a change in cell dimensions (**b**) and crystal corner angle (**a**). Colored lines in the panels c and d correspond to a few individual molecules; one from each layer in the simulation cell. The black line follows the average over all 512 molecules (1024 dihedrals). The vertical dashed lines indicate the times at which the temperature in the simulation is changed from 27 °C via a linear ramp (between 60 and 160 ps) to 177 °C. The background colors visualize this temperature increase from blue to red. The outside colored short lines on the left and right borders of the graphs are the corresponding experimental values of the $\alpha$-phase (at 27 °C) and $\beta$-phase (at 177 °C), respectively. Cooling the thus obtained structure to 27 °C shows that the energy of this new form is higher than that of the $\alpha$-phase at the same temperature, making it indeed a reversible transition driven by entropy. Source data are provided as a Source Data file

(0.15 mg) of the glass bead, the kinetic energy generated by the deformation could be estimated to be $>6.5 \times 10^{-11}$ J. This amount of energy would allow the crystal, when standing on one of its side faces, to jump 3 mm high, which can indeed explain that our crystal can jump over a long distance. The work density (output mechanical energy per unit volume or mass of material) in this process is $>270$ J kg$^{-1}$, which is two orders of magnitude larger than the typical performance of most MEMS (microelectromechanical systems)[23,32] or the recently discovered autonomous actuator[33]. Actuators are typically characterized by strain[34] and strain rate, i.e. the change in shape per second[35]. From the time-dependent microscopy observations we can estimate that the largest strain rate for some micron-sized 4-DBpFO crystal is larger than 180% s$^{-1}$ (Supplementary Fig. 7), comparable with thermally activated shape memory alloys ($<300\%$ s$^{-1}$) but much larger than that of piezoelectric actuators ($<12\%$ s$^{-1}$)[23] (Supplementary Table 1). In fact, the strain rate was too large to be determined for many of the smaller, high quality crystals

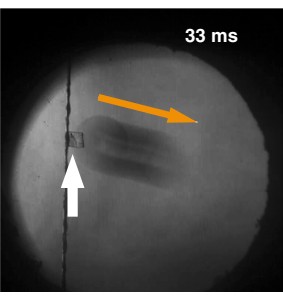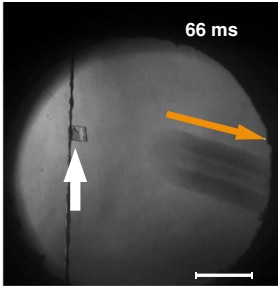

**Fig. 5** Glass bead displacement by crystal deformation. The force generated by the shape deformation was evaluated by the displacement of a 0.15 mg glass bead. White arrows highlight the position of the organic crystal, while orange ones mark the position of the bead at 0 ms and the direction of the bead movement at 33 and 66 ms, respectively. These pictures were extracted from the Supplementary Video 3. Scale bar is 0.5 mm

(that thus likely had fewer pinning dislocations or impurities), as their phase boundary can hardly be observed because of the high speed of the phase transition. To estimate the generated force during the phase transition, a crystal was placed between two parallel glass plates on a flat substrate. Here, the measured pushing force onto the glass plates by the crystal is equivalent to the maximum static friction between the glass plates and the substrate. A small crystal ($160 \times 160 \times 60$ μm) can generate a pushing force as large as 530 μN at its phase transition temperature, corresponding to $10^4$ times its gravitational force (Supplementary Fig. 8b and Movie 4). When the glass plates were too heavy, the molecular movement became distorted and incoherent phase boundaries appeared during the phase transition (Supplementary Fig. 8c and Movie 5).

As the crystal lattice completely returns to its initial state through the reversible phase transition, the "shear strain" of the crystal, which is as high as 0.18 during the transformation (Supplementary Fig. 4), is completely elastic. Although the optimum strain of some metals could be more than 0.10 in theory[36], the real maximum elastic strain only reaches 0.01 in most polycrystalline and even nanocrystalline alloys[37] due to the defects in the crystals[38]. Moreover, elastically deformable metals can only generate unidirectional forces—when the deformed lattice returns to its stable state. Here, the lattices of both phases are stable, so both the heating and cooling induced shape change can be exploited to generate a force. That is, the lattice change during heating is endothermic, converting thermal energy into kinetic energy while storing thermal energy in the crystal in the form of chemical energy. This stored chemical energy is subsequently released and converted into kinetic energy during cooling, due to the reversible shape change.

## Discussion

In summary, we have fabricated an organic thermosalient crystal 4-DBpFO that can be applied as a molecular microactuator. The crystal shows a reversibly major shear deformation in the **(010)** plane, which proceeds along two orthogonal crystal sides with excellent reversibility for more than 100 cycles of the temperature induced structural phase transition without visible wear. The applicability of this controllable and strong deformation was demonstrated by its ability to displace a load of more than 100 times its own mass for at least 5 times. The remarkable reversibility and robustness of this transition arises from in-plane molecular rotations through a nucleation-and-growth mechanism as demonstrated by Molecular Dynamics simulations. This is contrary to many cases in the literature where some thermosalient behavior is explained by cooperative motion on much larger length scales. This molecular structure can be used as a guidance to design new types of controllable and resilient self-assembly molecular microactuator that may operate in desired temperature ranges.

## Methods

**Materials**. The synthesis of the 4-DBpFO compound can be found in reference [15]. The α-phase crystal was grown through a solution-diffusion method. Heptane was dropped gently on the top of a saturated solution of chloroform, and then heptane will diffuse into the chloroform slowly. Well-defined parallelogram-shaped microcrystals with side lengths varying from 2 to 50 μm were formed after 24 h. After several weeks of diffusion, larger single crystals with side lengths ranging from 100 to 200 μm could be obtained, suitable for single crystal XRD analysis.

**Optical microscopy**. Fluorescence images were acquired with a Leica-Microsystems DM2500 microscope. Optical microscope images were obtained with a Zeiss Axioplan 2 microscope, equipped with a MediaCybernetics Evolution VF digital camera. For images of the shape change this microscope was equipped with a Linkam LTS420 thermal stage under nitrogen atmosphere, using heating rates ranging from 3 to 10 °C min$^{-1}$. The movie of the "kicking" glass bead was taken with a Leica WILD M10 microscope, equipped with an Evolution MP5.0 Leica DMC2900 digital camera with frame rate of 30 fps and a Linkam LTS420 thermal stage under nitrogen atmosphere.

**Differential scanning calorimetry**. Differential scanning calorimetry (DSC) measurements were carried out with a Mettler Toledo DSC1 calorimeter equipped with a high sensitivity sensor (HSS8), in combination with LN2 liquid nitrogen cooling, a sample robot and STAR$^e$ software 13.00a. As-grown microcrystals (about 0.45 mg) were heated and cooled in a sealed aluminum pan with a rate of 10 K min$^{-1}$ in the temperature range of 50–205 °C. The heat flow was measured by comparing to an empty reference pan as a function of temperature. The calorimeter was calibrated with the melting points of indium ($T_{In} = 429.5$ K and $\Delta H = -28.13$ J g$^{-1}$) and zinc ($T_{Zn} = 692.85$ K and $\Delta H = -104.77$ J g$^{-1}$), both supplied by Mettler Toledo. The enthalpy change of the phase transition was calculated by integrating the exothermic peak and the endothermic peak, respectively.

**Single crystal X-ray diffraction**. Single crystal X-ray diffraction (SXRD) measurements were collected on a Bruker D8 Quest diffractometer with sealed tube (MoKa radiation) and Triumph monochromator ($\lambda = 0.71073$ Å). The software package Saint was used for the intensity integration[39]. Absorption correction was performed with SADABS[40]. The structures were solved with direct methods using SHELXT[41]. Least-squares refinement was performed with SHELXL-2014[42] against $\left|F_h^o\right|^2$ of all reflections. Non-hydrogen atoms were refined freely with anisotropic displacement parameters. Hydrogen atoms were placed on calculated positions or located in difference Fourier maps. All calculated hydrogen atoms were refined with a riding model. For high temperature SXRD, the crystal was sealed in a capillary and an Oxford Cryostream 700 plus was used to heat the crystal to the desired temperature.

**Molecular dynamic simulation method**. Molecular dynamic (MD) simulations were performed using LAMMPS[43] (version 16-02-2016). A simulation cell was build out of $8 \times 2 \times 8$ crystallographic unit cells of the α-phase (512 molecules). The GAFF2.1 force field was used for the intra- and intermolecular interactions[44] in combination with ESP-derived point charges (DFT, B3LYB cc-pVDZ). The latter were calculated with the help of Molpro and Molden[45,46]. Two modifications were applied to the GAFF2.1 force field. The contribution for the torsional change in dihedral θ was made asymmetric in accordance with DFT calculations, and the H atom sites were chosen between the nucleus and the maximum in the electron density as determined by XRD. For all interactions, the dispersion contribution is cut-off at 10.0 Å and the Coulombic part of the potential is computed using Ewald summation with a relative RMS error in per-atom force of $10^{-6}$. The simulations were performed in the triclinic NPT ensemble with a time step of 0.5 fs. The thermostat and barostat parameters were set to 40 and 400 fs, respectively.

**Estimation of actuator performance.** *Strain:* Crystal strain at the phase transition temperatures: The strain during the phase transition was calculated from the lattice parameters of the two phases. The linear strain can be calculated from:

$$\varepsilon = \frac{\Delta l}{l} \qquad (1)$$

where '*l*' is the cell length, and "$\Delta l$" is the corresponding expansion length along the cell axes after the phase transition. An elastic compressive strain of 0.094 along the *a*-axis and an elastic tensile strain of 0.069 along the *c*-axis can be reached during the phase transition from the *α*-phase to the *β*-phase by heating, while an elastic tensile strain as large as 0.10 along the *a*-axis and an elastic compressive strain of 0.064 along the *c*-axis can be reached during the reversed phase transition by cooling. In engineering, shear strain is equal to the length of deformation at its maximum divided by the perpendicular length in the plane of force application. By this definition, the shear strain here is calculated from:

$$\gamma = \frac{\Delta x}{y} \qquad (2)$$

where "$\Delta x$" is the displacement along the lattice diagonal and *y* is the cell length perpendicular to the displacement direction (Supplementary Fig. 4). The shear strain was calculated to be as large as 0.18.

*Kinetic energy:* The capability of delivering kinetic energy of the crystals was estimated by moving a micron-sized glass bead. The mass of the crystal shown in Fig. 5 and in Supplementary Movie 3 was ~2.5 µg, which was calculated from the relative molecular mass and the size of the crystal ($200 \times 200 \times 50$ µm). The bead moved over 0.96 mm in 0.033 s, which corresponds to an average speed of 29 mm s$^{-1}$. The kinetic energy ($6.5 \times 10^{-11}$ J) of the bead obtained from the shape change of the crystal was calculated from:

$$E_k = \frac{1}{2} m v^2 \qquad (3)$$

where "*m*" is the mass of the bead (0.15 mg), and "*v*" is the speed of the bead induced by the movement of the crystal.

*Maximum pushing force:* The maximum pushing force of the microcrystals was estimated by the ability of moving glass plates (as shown in Supplementary Fig. 8). Crystals were placed between two parallel glass plates on a larger flat glass substrate. The parallel glass plates can be pushed apart by the crystals at the phase transition temperature, due to the shape change of the crystals. The force acting on the glass plates generated by the shape change (*F*) was calculated from:

$$F = f_s^{max} = \mu_s m g \qquad (4)$$

where "$f_s^{max}$" is the maximum static friction between the glass plates and the glass substrate; "$\mu_s$" is the static friction coefficient between two clean and dry glass surfaces, which is 0.9–1.0;[47] "*m*" is the mass of a single glass plate and "*g*" is the gravity acceleration. The glass plates in Supplementary Fig. 8b each weigh 60 mg, which is ~$3 \times 10^4$ times heavier than the crystal ($160 \times 160 \times 60$ µm), so the corresponding static friction between the glass plates and the glass substrate is at least $0.9 \times 60$ mg $\times$ 9.8 N kg$^{-1} \approx 530$ µN, which is ~$3 \times 10^4$ times stronger than its gravitational force. When each glass plate weighs 260 mg, which is ~$10^5$ heavier than the crystal ($200 \times 200 \times 50$ µm), the glass plates can still be pushed apart by the crystal, but the crystal becomes "damaged" during the shape change (Supplementary Fig. 8c). During this process, the static friction between the glass plates and substrate can be as large as 2300 µN, which is $10^5$ times the gravitational force of the crystal.

## Data availability

Crystal data of the *α*-phase and *β*-phase are available from the Cambridge Crystallographic Data Centre with reference number of CCDC-1862516 and CCDC-1920480, respectively. The source data underlying Figs. 3a and 4 are provided as a Source Data file. All data needed to evaluate the conclusions in the paper are present. Additional data related to this paper may be requested from the authors.

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

## Acknowledgements

We thank K. Li from the Center for High Pressure Science and Technology Advanced Research (HPSTAR) in Beijing for help with the high temperature SXRD, and T. Toonen and E. Ronde from Radboud University for their technical supports. Y. D gratefully acknowledges the financial supports from the China Scholarship Council. Part of this work was supported by the Netherlands Organization of Scientific Research (NWO SPI63.256) and National Natural Science Foundation of China (21773168). Beamline BL17B of National Facility for Protein Science Shanghai (NFPS) at Shanghai Synchrotron Radiation Facility (SSRF) is acknowledged for providing assistance during data collection.

## Author contributions

Y.D., J.X., S.S., and T.R conceived the study. T.R, H.C., and J.X supervised the work. Y.D. carried out the experimental studies. P.T performed the SXRD measurements. H.C performed the MD simulation. The paper was initially drafted by Y.D. and T.R., and the molecular dynamic simulation was drafted by H.C with support from J.X, S.S, and P.T. All authors made comments in editing of the final paper and Supplementary material.

## Competing interests

The authors declare no competing interests.
