## [Peer Review File · Nature Communications]

Reviewers' comments:

Reviewer #1 (Remarks to the Author):

I read with interest this manuscript which reports the phase transition of a crystal of a fluorenone derivative which goes through a thermal transition by deformation. I must admit I was rather underwhelmed by the conclusions, which, although the title promises much, seem to deliver little. There is a great attention recently in molecular crystals that can be mechanically deformed when they are affected by heat or light. The results described in this work are another example in the series of reports of such effects, where a phase transition can display a mechanical response. These materials are thought of new materials, although some of the changes have been reported before. The work presented here is a thorough, but it is not the first in the line. Similar effects in superelastic and superplastic crystals were described by Takamizawa et al. in several articles on several compounds recently, and some of the systems show reversibility similar or better than the one described. The thermosalient effect was studied by Naumov et al. and there are many examples reported. The change in the angles during irradiation of diarylethene crystals were reported in Nature by Irie et al., and they are comparable to the one described here. Finally, the martensitic transitions in reversibly deforming crystals were recently reported by Diao et al. It looks like this work is a combination of these previously already reported effects, supplemented by calculations and simple demonstration of the actuation, similar to the one reported by Irie and Morimoto for bending crystal which can lift a ball or actuate a gear. The mechanism of the transition was described in detail, however in my opinion the material does not bring to a significant advantage compared to the previously reported materials which were reported to be reversible and very fast.

There are a few other aspects of this work that might be problematic. It was described that the phase transition rate can be controlled by temperature. This is not a virtue for thermal actuation, which requires very fast deformation at a minimal change of temperature. Devices based on martensitic transitions require very fast transition with rate that does not depend on temperature. Secondly, the change in shape described are very small. For example, Tao et al. have described materials which can expand over 100% during a phase transition. Such transitions with large change in dimensions will produce more work. Third, the calculation of the actuation energy probably should be corrected to take into consideration factors such as drag and friction. Lastly, the actuation performance is compared with individual examples. The material should be compared to other materials, both molecular crystals and others. Considering all this, I don't find the novelty of this manuscript sufficient for publication in Nature Communications. The work may become publishable after some modification in a more specialized journal, such as Crystal Growth and Design or CrystEngComm.

Reviewer #2 (Remarks to the Author):

I think that this paper could be published on Nature Communications after revision.

I am raising a few essential points for improving the paper.

1. In the latter parts of paper, the observed value estimated by mechanical works should be within a large allowance according to the description that the velocity in thermal phase transition is hard to be controlled. Please make the availability of the "representative value" clear by showing the distribution of the values with the statistical treatment.

2. The inhibition of the proceeding of phase transition under mechanical stress (The observation is showed in Figure S7) indicates the generation of organosuperelasticity (or organoferroelasticity) and/or shape-memory effect with organosuperelasticity. The description have been missed.

Reviewer #3 (Remarks to the Author):

In recent years, dynamic molecular crystals attracted a significant attention for their potential applications as actuators, and rightfully so – these materials are generally softer than the conventional hard actuators (e.g. ceramics), and at the same time harder than soft actuators (e.g. elastomers). Therefore, they could potentially fill the “uncharted” middle part of the elasticity spectrum, and find applications in soft robotics. Moreover, their crystalline order allows for fast energy transfer, and could elicit super-fast response time. At the same time, the crystalline order allows for structural characterization, and ultimately better understanding of the underlying physics.

Duan et al. report on a thermo-elastic micro-actuator based on an organic molecular crystal. The material is made of a fluorenone center and two phenyl rings at each side linked by single bonds. Basically, the molecule is made of five flat segments, linked by flexible single bonds, that allows for flexibility and dynamics in the solid state. In fact, the authors show that this material is thermosalient (exhibits thermo-induced jumping), and it is quite different in structure and performance than hitherto published thermosalient crystals. The authors studied this material with single crystal diffraction and molecular dynamics and found that small conformation changes elicit large change of the macroscopic shape of the crystal. Moreover, they successfully managed to build a crystalline machinery and repeatedly displace a micron-sized glass bead, with work density of 270 J/kg.

These results are impressive. However, there are several aspects that has not been addressed. I would recommend this work to be published in Nature Communications, after the following major comment, and several minor comments are resolved or explained:

Major comment:

In order to explain the thermosalient effect, knowledge on the crystal structure before and after the transition is more than necessary! The authors say that during the phase transition single crystals delaminate, thwarting structure solutions. However, single crystal diffraction is not the only way to solve a crystal structure. The fact that the authors managed to get the unit cell and possibly the symmetry of the structure, and the fact that transition is perfectly reversible, is a strong indication that the sample is in fact (micro)crystalline and it can be analyzed by powder diffraction. Crystal structure from powder diffraction data followed by a Rietveld refinement can provide detailed crystal structure, even using a laboratory powder diffractometer. I strongly believe that this kind of study cannot be published without knowing the crystal structure of the high-temperature phase, especially not in a highly reputable journal such as Nature Communications.

Minor comments:

- 1) The right format of the space groups (Hermann-Mauguin notation) is roman font for every number and italic for every letter. That should be corrected throughout the manuscript. (For example in lines 69, 71, 104, 105, and so on.)
- 2) Standard deviations of the unit cell parameters, bond lengths and angles must be provided (For example lines 74, 105-107 and so on.)
- 3) The Miler Index notation should be corrected. $\{hkl\}$ denotes all planes that are equivalent to the (hkl) by the symmetry of the crystal. For example, in the caption of Fig. 1. should be corrected that the molecular packing is shown in the (010) and (001) planes.

Reviewer #4 (Remarks to the Author):

This study dedicated to analysis of reversible phase transition in organic molecular crystal on heating accompanied with strong mechanical response and shape change. Mechanically responsive materials are of great interest for materials science, chemistry and physics. The topic of the paper shall attract attention of the most readers of the journal. At the same time some aspects of the paper are ambiguous and require further explanation and/or clarification. Therefore, paper publication is possible only after major revision. Please, find further remarks and comments below.

1) Planes and directions should be denoted as (010) and [010] respectively (instead of {010} and <010>).

2) Line 91. It is not completely clear why authors call shape of α -phase as «pseudo-rhombic» although it is called «parallelogram-like» at line 75.

3) Line 174. Statements «almost the same» and «is almost invariant plane» are not appropriate for research paper. Procedure of finding invariant plane is described in {Bhadeshia, H. K. D. H. (2006). Worked Examples in the Geometry of Crystals. London: Institute of Metals} that allows more precise determination to prove martensitic nature of this phase transition.

4) Representation of MD simulations and Movie S2 itself are not informative. In general case conformational changes as well as changes in lattice parameters do not necessarily cause crystal shape deformation. (010) plane where most of changes occur is also not visible in Movie S2 since it is perpendicular to the screen.

5) Despite authors did not solve crystal structure of high-temperature β -phase due to damage of the crystals large enough for X-ray diffraction experiment, it is still possible to propose the crystal structure model of martensitic transformation product basing on initial crystal structure of α -phase and optical microscopy data for the phase transition. The correct procedure is described in (<http://journals.iucr.org/m/issues/2017/05/00/lt5002/index.html>). Adding the model based on optical microscopy observations could additionally confirm MD results.

6) Line 255. 29 mm/s should probably be an average velocity but not «minimum».

7) Line 268. Phase boundary can be overlooked due to higher speed of phase transition. That does not mean that it is absent. This part needs to be re-worded.

8) Line 308. There is no any contraries to the general consensus in the literature since the phase transition described in this contribution may not be necessarily thermosalient. The term «thermosalient» is fair for thermally induced jumping or leaping crystals ([dx.doi.org/10.1021/acs.chemrev.5b00398](https://doi.org/10.1021/acs.chemrev.5b00398) and references therein) where strain is accumulated during the induction period and is released suddenly leading to crystal self-actuation (!). Crystal self-actuation is also strongly dependent on phase transition speed that can be varied by thermal ramp rate in this case. Moreover, any first order solid state phase transition requires nucleation with subsequent growth of new phase that occurs in many thermosalient materials. Please, re-phrase this part of the text.

Response to comments Reviewers on manuscript entitled "Robust thermo-elastic micro-actuator based on an organic molecular crystal" (NCOMMS-19-04545).

Reviewer #1:

I read with interest this manuscript which reports the phase transition of a crystal of a fluorenone derivative which goes through a thermal transition by deformation. I must admit I was rather underwhelmed by the conclusions, which, although the title promises much, seem to deliver little.

Reply 1: We thank the reviewer for her/his criticism on our manuscript. However, we do not understand the remark about delivering little: due to its large strain and strain rate, the capability of the crystal actuator described here is not little compared with that of reported artificial actuators (see Table S1). While its work density is comparable with biological muscles, it is 10^6 higher than the artificial actuators based on Shape-Memory Alloys or Piezoelectric actuators.

There is a great attention recently in molecular crystals that can be mechanically deformed when they are affected by heat or light. The results described in this work are another example in the series of reports of such effects, where a phase transition can display a mechanical response. These materials are thought of new materials, although some of the changes have been reported before. The work presented here is a thorough, but it is not the first in the line. Similar effects in superelastic and superplastic crystals were described by Takamizawa et al. in several articles on several compounds recently, and some of the systems show reversibility similar or better than the one described.

Reply 2: We thank the reviewer for bringing this up, so we can clarify this. Indeed, Takamizawa et al. have reported superelastic organic crystals with good reversibility (*Nature Commun.* 2018, 9, 3984; *Angew. Chem. Int. Ed.* 2014, 53, 6970-6973, etc.). However, the mechanism of their crystals is very different for the crystals we report here. Although both cases are based on single-crystal-to-single crystal phase transitions, the superelasticity we report here results from a temperature change (thermal elastic), while the phase transitions reported by Takamizawa et al. are mainly induced by applying a stress on the crystals (mechanical elastic). A thermal elastic phase transition as reported by us allows to convert thermal energy into kinetic energy, which can be applied in devices such as a thermal actuator.

We have highlighted the contributions of Takamizawa et al. (line 264-265 on page 10), and have cited the relevant literatures (Refs. 14 in the reference list).

The thermosalient effect was studied by Naumov et al. and there are many examples reported.

Reply 3: The reviewer is right in that the thermosalient effect has indeed been well studied by Naumov et al. (*Chem. Rev.* 2015, 115, 12440-12490; *J. Am. Chem. Soc.* 2013, 135, 12241-12251; *Nat. Commun.* 2014, 5, 4811; as we have cited in Refs. 1, 2, and 3). However, in most of these crystals, the thermosalient effect cannot be controlled, and the crystals usually jump randomly upon heating without demonstrating an effective mechanical output. Here, we demonstrate the use of a thermosalient organic crystal as a micro-crystalline machine that achieves an effective mechanical output. More importantly, the path of the shape change in our case is very different from the reported examples. While the

shape change in layered crystals is usually caused by shifting between layers, with a phase boundary parallel to the layer planes (*J. Am. Chem. Soc.* 2013, 135, 12241-12251; *J. Am. Chem. Soc.* 2016, 138, 13298-13306), the phase boundary in our crystal is perpendicular to the layer planes. This represents a new class of thermosalient crystals which could be used as a guidance to design new types of controllable and resilient self-assembly molecular micro-actuators.

The change in the angles during irradiation of diarylethene crystals were reported in Nature by Irie et al., and they are comparable to the one described here.

Reply 4: The corner angles of the diarylethene crystals reported by Irie *et al.* (*Nature* 2007, 446, 778-781, Ref. 11 in our manuscript), change from 88° and 92° to 82° and 98°, respectively, while for our crystal they change from 88° and 92° to 77° and 103°, respectively. The changes in the angles in our crystal are thus twice as large and the corresponding shear strain is therefore also two times larger, which is not 'comparable' for a shear deformation.

Finally, the martensitic transitions in reversibly deforming crystals were recently reported by Diao et al.

Reply 5: Diao *et al.* indeed reported reversibly deforming crystals similar to the Martensitic phase transition (*Nature Commun.* 2018, 9, 278, added as a new reference in Ref. 5 in our manuscript), but to characterize a martensitic phase transition, one should analyze it from a crystallographic point of view, showing the lattice relationship between the two phases (Porter, D. A., et al. *Phase Transformations in Metals and Alloys* (CRC Press, 2009), as we have done. We also show that the (101) and (10-1) planes in our crystal are invariant during the phase transition, so no additional lattice invariant shear is needed to keep the invariance of the phase boundary. This is also one of the reasons why the phase transition shows excellent reversibility. In principle, organic crystals easily break if an additional lattice invariant shear is present. Moreover, we are the first to report that the phase transition in an organic crystal can proceed via two different paths, resulting in the same final crystal shape but in a mirrored orientation (line 136-140, page 6), which further confirms the martensitic feature.

It looks like this work is a combination of these previously already reported effects, supplemented by calculations and simple demonstration of the actuation, similar to the one reported by Irie and Morimoto for bending crystal which can lift a ball or actuate a gear. The mechanism of the transition was described in detail, however in my opinion the material does not bring to a significant advantage compared to the previously reported materials which were reported to be reversible and very fast.

Reply 6: Our crystals demonstrate many advantages compared with the previously reported crystals, such as larger shear strain and robustness, and pronounced dimensional changes in two directions instead of one: in one direction it expands and in the other it shrinks. Our crystals therefore are not only useful as micro-actuators but also perfectly meet the conditions to design thermal contraction materials described by Miller *et al.* (*Nature Mater.* 2010, 9, 7-8). Moreover, the two different shear directions can be exploited to design more complex thermally controlled switches, as described by Diao *et al.* (*Nature Commun.* 2018, 9, 278).

There are a few other aspects of this work that might be problematic. It was described that the phase transition rate can be controlled by temperature. This is not a virtue for thermal actuation, which requires very fast deformation at a minimal change of temperature. Devices based on martensitic transitions require very fast transition with rate that does not depend on temperature.

Reply 7: We thank the reviewer for raising this issue. We describe in our article that the strain rate (phase transition speed) may depend on the crystal quality. For the smaller, higher quality crystals, the phase transition speed is too high to follow the phase boundary by our high speed camera, which means a very fast deformation. We have clarified this in the text (Line 253-256, page 9-10).

Secondly, the change in shape described are very small. For example, Tao et al. have described materials which can expand over 100% during a phase transition. Such transitions with large change in dimensions will produce more work.

Reply 8: The change described here (shear strain of 18%) is one order of magnitude higher than that in the typical PHA crystals (strain of 1.6%) (*J. Am. Chem. Soc.* 2019, 141, 3371-3375, *Nature Commun.* 2014, 5, 4811) and three times that of the $[\text{Ni}^{\text{II}}(\text{en})_3](\text{ox})$ complex (strain of 5%) (*Nat. Chem.* 2014, 6, 1079-1083). Moreover, a large shape change does not necessarily mean a large strain rate. Only crystals show both considerable shape change as well as strain rate and thus can produce more work.

Third, the calculation of the actuation energy probably should be corrected to take into consideration factors such as drag and friction.

Reply 9: The reviewer is right that to calculate the actuation energy accurately, it is important to take into consideration factors such as drag and friction. However, it is very difficult to consider these factors in the kicking glass bead experiment. Furthermore, if the factors such as drag and friction are taken into account, the energy density will be even larger than the value that we reported here. In addition, the maximum force is estimated by calculating the maximum static friction between the glass sheet and the substrate, and there the friction has been considered.

Lastly, the actuation performance is compared with individual examples. The material should be compared to other materials, both molecular crystals and others.

Reply 10: We have compared the performance with that of crystalline diarylethene, biological muscles, piezoelectric actuators, shape memory alloys, and other thermosalient crystals. A supplementary table (Table S1 in the Supplementary Information) is added to summarize some of these performances. The work density of the 4-DBpFO crystalline actuator is 10^6 higher than that of artificial actuators based on Shape-Memory Alloy or Piezoelectric actuators. We have not found work density values for other thermosalient crystals, but the strain of the 4-DBpFO crystals is much larger than for most reported thermosalient crystals.

Considering all this, I don't find the novelty of this manuscript sufficient for publication in Nature Communications. The work may become publishable after some modification in a more specialized journal, such as Crystal Growth and Design or CrystEngComm.

Reply 11: As we have answered all the comments of this reviewer and also those from the other 3 reviewers, we hope she/he will reconsider this opinion.

Reviewer #2:

I think that this paper could be published on Nature Communications after revision. I am raising a few essential points for improving the paper.

1. In the latter parts of paper, the observed value estimated by mechanical works should be within a large allowance according to the description that the velocity in thermal phase transition is hard to be controlled. Please make the availability of the “representative value” clear by showing the distribution of the values with the statistical treatment.

Reply 1: We thank the reviewer for the positive comments. We agree that the observed value estimated by mechanical work is within a large allowance. This is not only because the phase transition is hard to control, but also the crystal quality plays an important role. Moreover, a little difference in the relative position between the crystal and the glass bead can also result in a large allowance of the mechanical works. Therefore, we gave a conservative value for the energy density. Nevertheless, as the performance of the actuator here is mainly determined by its strain and strain rate, and the strain is the same for different crystals, we can estimate the general performance of the crystals by showing the distribution of the strain rate. A supplementary figure has been added in the Supplementary Information (see Figure. S8) to show this. On the other hand, the force generated by an elastic shape change only relates to the strain, not the strain rate, according to Hooke's law. Therefore, the force, which is estimated to be 530 μN , is not related to the phase transition speed.

2. The inhibition of the proceeding of phase transition under mechanical stress (The observation is showed in Figure S7) indicates the generation of organosuperelasticity (or organoferroelasticity) and/or shape-memory effect with organosuperelasticity. The description have been missed.

Reply 2: Thank you for pointing this out. Organosuperelasticity, as studied by Takamizawa *et al.*, refers to a shape change by applying a stress, while the shape can recover after removing the stress. However, we do not observe any shape change by applying stress to the 4-DBPFO crystal. The shape change in Figure S9 (former S7) only results from the temperature change and is not caused by stress. However, the crystal can indeed have a shape-memory possibility. As shown in the main text, in some crystals the phase boundary can be pinned at a certain position (probably a defect place) by keeping them at a constant temperature, and its propagation direction could be even reversed upon cooling (Line 132-137, page 6). This means that the shape can be changed by tuning the temperature. In other words, the crystal studied here indeed shows organosuperelasticity, induced by temperature, not stress. We have included this description (Supplementary Figure S9), highlighted the contributions of Takamizawa *et al.* in the field of organosuperelasticity (line 264-265 on page 10), and have cited the relevant literatures (Refs. 14 in the reference list).

Reviewer #3:

In recent years, dynamic molecular crystals attracted a significant attention for their potential applications as actuators, and rightfully so – these materials are generally softer

than the conventional hard actuators (e.g. ceramics), and at the same time harder than soft actuators (e.g. elastomers). Therefore, they could potentially fill the “uncharted” middle part of the elasticity spectrum, and find applications in soft robotics. Moreover, their crystalline order allows for fast energy transfer, and could elicit super-fast response time. At the same time, the crystalline order allows for structural characterization, and ultimately better understanding of the underlying physics.

Duan et al. report on a thermo-elastic micro-actuator based on an organic molecular crystal. The material is made of a fluorenone center and two phenyl rings at each side linked by single bonds. Basically, the molecule is made of five flat segments, linked by flexible single bonds, that allows for flexibility and dynamics in the solid state. In fact, the authors show that this material is thermosalient (exhibits thermo-induced jumping), and it is quite different in structure and performance than published thermosalient crystals. The authors studied this material with single crystal diffraction and molecular dynamics and found that small conformation changes elicit large change of the macroscopic shape of the crystal. Moreover, they successfully managed to build a crystalline machinery and repeatedly displace a micron-sized glass bead, with work density of 270 J/kg.

These results are impressive. However, there are several aspects that has not been addressed.

I would recommend this work to be published in Nature Communications, after the following major comment, and several minor comments are resolved or explained:

Reply 1: We thank the reviewer for the positive comments.

Major comment:

In order to explain the thermosalient effect, knowledge on the crystal structure before and after the transition is more than necessary! The authors say that during the phase transition single crystals delaminate, thwarting structure solutions. However, single crystal diffraction is not the only way to solve a crystal structure. The fact that the authors managed to get the unit cell and possibly the symmetry of the structure, and the fact that transition is perfectly reversible, is a strong indication that the sample is in fact (micro)crystalline and it can be analyzed by powder diffraction. Crystal structure from powder diffraction data followed by a Rietveld refinement can provide detailed crystal structure, even using a laboratory powder diffractometer.

I strongly believe that this kind of study cannot be published without knowing the crystal structure of the high-temperature phase, especially not in a highly reputable journal such as Nature Communications.

Reply 2: Thank you very much for this comment that forced us to obtain this structure after all. We finally got the high-temperature structure through single crystal X-ray diffraction using high quality crystals which did not delaminate during the phase transition (See the following figure). This crystal structure is reproduced in different crystals, so it is highly reliable. We have submitted this structure to the Cambridge Crystallographic Data Centre with reference number of CCDC-1920480.

The high temperature beta phase has space group $P2_1/n$ (in the main text, we use a non-standard description $P2_1/n$ 1 1 unique axis-a), with a monoclinic angle deviating slightly from 90° . The cell lengths are the same as with the previously suggested structure with space group $Pmc2_1$. We further used nonlinear optical measurements to confirm this space group (Fig. S7b). There is no second harmonic generation from the crystals after the phase transition, which confirms that the β -phase belongs to the centrosymmetric $P2_1/n$ and not the noncentrosymmetric $Pmc2_1$.

The proposed phase transition mechanism is further supported by this new structure. The unit cell parameter changes of the (010) plane at the phase transition point are consistent with the crystal shape change observed in optical microscopy. The shape change is caused by the in-plane anisotropic lattice expansion, according to the mechanism proposed in the manuscript. Moreover, the new structure also suggests that the phase transition mainly results from the in-plane rotation of the benzene rings, while the relative position of the fluorenone center does not change. This is also consistent with our molecular dynamic simulations.

We have revised the text according to the single crystal structure of the high temperature phase (Line 89-99, page 4, Fig. 1 in page 3 and Table.1 in page 4).

Minor comments:

1) The right format of the space groups (Hermann-Mauguin notation) is roman font for every number and italic for every letter. That should be corrected throughout the manuscript. (For example in lines 69, 71, 104, 105, and so on.)

Reply 3: We are sorry for the wrong fonts. We have corrected this according to the comment.

2) Standard deviations of the unit cell parameters, bond lengths and angles must be provided (For example lines 74, 105-107 and so on.).

Reply 4: Thanks. We have added the standard deviations of the unit cell parameters of both phases (Table.1 in page 4)

3) The Miler Index notation should be corrected. $\{hkl\}$ denotes all planes that are equivalent to the (hkl) by the symmetry of the crystal. For example, in the caption of Fig. 1. should be corrected that the molecular packing is shown in the (010) and (001) planes.

Reply 5: We are sorry for the mistakes in the Miler Index notation. We have checked through the manuscript and made corrections where necessary.

Reviewer #4:

This study dedicated to analysis of reversible phase transition in organic molecular crystal on heating accompanied with strong mechanical response and shape change. Mechanically responsive materials are of great interest for materials science, chemistry and physics. The topic of the paper shall attract attention of the most readers of the journal.

Reply 1: We thank the reviewer for the very positive comments.

At the same time some aspects of the paper are ambiguous and require further explanation and/or clarification. Therefore, paper publication is possible only after major revision. Please, find further remarks and comments below.

1) Planes and directions should be denoted as (010) and [010] respectively (instead of {010} and <010>).

Reply 2: We are sorry for the mistakes in the notation. We have corrected all parentheses and curly brackets to match the IUCr standard.

According to the book of (Th. Hahn, International Tables for Crystallography (2006). Vol. A, ch. 1.1, pp. 2-3):

(hkl) - Indices of a crystal face, or of a single net plane (Miller indices);
{hkl} - Indices of a set of all symmetrically equivalent crystal faces ('crystal form'), or net planes;

[uvw] - Indices of a lattice direction (zone axis);

<uvw> - Indices of a set of all symmetrically equivalent lattice directions.

2) Line 91. It is not completely clear why authors call shape of α -phase as «pseudo-rhombic» although it is called «parallelogram-like» at line 75.

Reply 3: We have changed "pseudo-rhombic" to "parallelogram-like". (Line 84)

3) Line 174. Statements «almost the same» and «is almost invariant plane» are not appropriate for research paper. Procedure of finding invariant plane is described in {Bhadeshia, H. K. D. H. (2006). Worked Examples in the Geometry of Crystals. London: Institute of Metals} that allows more precise determination to prove martensitic nature of this phase transition.

Reply 4: We are sorry for the confusing description. The (101) or (10-1) plane is indeed an invariant plane during the phase transition. As we have shown in Fig. 2f, Fig S4 and Table S1, although the in-plane c -axis changed from 7.27 to 7.7 Å, and a -axis changed from 6.9 to 6.25 Å, the diagonal of the in-plan (010) plane lattice stays unchanged (10 Å) during the phase transition (from 170 °C to 178 °C), as

$$\sqrt{a_{170^\circ\text{C}}^2 + c_{170^\circ\text{C}}^2} = \sqrt{a_{178^\circ\text{C}}^2 + c_{178^\circ\text{C}}^2} = 10 \text{ \AA} .$$

Moreover, the b -axis also stays unchanged (52 Å) during the phase transition. In this way, the (101) or (10-1) planes are indeed invariant planes. This is consistent with that observed in optical microscopy: the phase boundary (invariant plane) is straight and perpendicular to the (010) plane. We have clarified this in the new manuscript (See Line 108-112, page 4 and supplementary Fig. S4).

4) Representation of MD simulations and Movie S2 itself are not informative. In general case conformational changes as well as changes in lattice parameters do not necessarily cause crystal shape deformation. (010) plane where most of changes occur is also not visible in Movie S2 since it is perpendicular to the screen.

Reply 5: Thanks for the comment. We have added a new Movie, in which the changes are more visible (Movie S2).

5) Despite authors did not solve crystal structure of high-temperature β -phase due to damage of the crystals large enough for X-ray diffraction experiment, it is still possible to propose the crystal structure model of martensitic transformation product basing on initial crystal structure of α -phase and optical microscopy data for the phase transition. The correct procedure is described in (<http://journals.iucr.org/m/issues/2017/05/00/lt5002/index.html>). Adding the model based on optical microscopy observations could additionally confirm MD results.

Reply 6: We finally got the high-temperature structure through single crystal X-ray diffraction using high quality crystals which did not delaminate during the phase transition (Fig.1, see also reply 2 to reviewer 3 above). We have submitted this structure to the Cambridge Crystallographic Data Centre with reference number of CCDC-1920480. We have also used the reviewer's suggested method (Ref. 21) to confirm this crystal structure (Fig. S1 and S7) and also used nonlinear optical measurements to confirm the centrosymmetric properties of the crystal after the phase transition (Fig. S7b). We have revised the text accordingly (Line 96-97, page 3 and Line 107-109, page 4).

6) Line 255. 29 mm/s should probably be an average velocity but not «minimum».

Reply 7: That is correct, and we have modified the manuscript accordingly.

7) Line 268. Phase boundary can be overlooked due to higher speed of phase transition. That does not mean that it is absent. This part needs to be re-worded.

Reply 8: We do agree with this comment, and modified this sentence accordingly: "In fact, the strain rate was too large to be determined for many of the smaller, high quality crystals (with thus likely fewer pinning dislocations or impurities), as their phase boundary cannot be followed because of the high speed of the phase transition" (Line 253-256, Page 9-10).

8) Line 308. There is no any contraries to the general consensus in the literature since the phase transition described in this contribution may not be necessarily thermosalient. The term «thermosalient» is fair for thermally induced jumping or leaping crystals ([dx.doi.org/10.1021/acs.chemrev.5b00398](https://doi.org/10.1021/acs.chemrev.5b00398) and references therein) where strain is accumulated during the induction period and is released suddenly leading to crystal self-actuation (!). Crystal self-actuation is also strongly dependent on phase transition speed that can be varied by thermal ramp rate in this case. Moreover, any first order solid state phase transition requires nucleation with subsequent growth of new phase that occurs in many thermosalient materials. Please, re-phrase this part of the text.

Reply 9: We agree with the reviewer and have changed this statement (Line 293-295, Page 11).

Reviewers' comments:

Reviewer #1 (Remarks to the Author):

In the first revised version of the manuscript as well as in their response to the reviewers' comments, the authors have provided stronger claims in support of the uniqueness of their material, and I am now convinced that this work warrants publication in Nature Communications. The combination of a thermosalient effect (which needs to be discussed in more detail), the robustness upon phase transition, the elegant molecular dynamics simulations, and the demonstration of application of this material together are important and interesting, and I believe that after another round of revision this manuscript will be acceptable for publication. In the revised version, the main claims were strengthened significantly by including the crystal structures of the two phases, and the mechanism of the phase transition is now much clearer. The reference list is now also more inclusive, and except for a couple of very recent publications (see my specific comments provided below), it reflects better the state of the art in this research. The authors may consider the following comments to improve their manuscript further. I also noticed that in their response to the first set of comments, the authors did respond however they did not make all of the changes in the manuscript; the authors are strongly encouraged to respond to the comments by making actual changes in the manuscript for each and every comment in addition to addressing the reviewers' comments in their response letter.

Most of the main text is dedicated to explanation of the mechanism of the phase transition, and that is definitely one of the most important aspects of this work. However, not much is said about the thermosalient effect per se. A reader who is not familiar with this topic might question whether this compound is really thermosalient material or not. The discussion is reduced to only one sentence stating one crystal jumps and a figure in the supplementary information. How do the crystals move? How many of them move and how many don't? With what speed and in which direction? How many retain their integrity and how many of them disintegrate? Please add a short discussion and include more details on the thermosalient effect at the beginning of the discussion section.

Introduction section, the first sentence: this sentence should be modified, because not all thermosalient transitions proceed in a single-crystal-to-single crystal manner.

The introduction, the sentence starting with "The transition appears to be triggered by small conformational changes in the molecules...". This goes well with some previous observations, and references with previous observations where the same conclusions were obtained should be added here.

Two other molecular crystals that were characterized as single crystal actuators were published recently (DOI: 10.1039/C9SC02444A, DOI: org/10.1021/jacs.8b12752). These articles are directly related to this work, and they should be included in the introduction and in the discussion. A comparison between the two materials and the one studied here should be added in the discussion of the main text, in the section "Shear deformation and durability".

In other thermosalient materials, usually the volume of the high-temperature phase is very slightly bigger (up to several percent) compared to that of the low-temperature one. In case of the material studied here, however, it looks like it is smaller. The authors should provide a brief explanation of this result and the reasons for the reduction of the volume.

This article reports a thermosalient effect, however the only figure that shows the effect is in the Supplementary material. The readers would benefit from a figure which shows the effect in the main text. This can be added to the current Figure 1.

Panel 1b: The contours, which represent a projection of the crystal shape, might be confused with

a unit cell. A contour of the actual unit cell should be added on the plots of both phases to distinguish it from the crystal shape.

Discussion section: The discovery of a new thermosalient compound which does not appear to belong to any of the three classes that were established previously by Sahoo et al., is a particularly novel and important aspect of this work, and this point of novelty is definitely something that brings this work within the scope of Nature Communications. The readers will benefit from a short discussion that explains the difference in the intermolecular interactions and molecular shape of this compound with other thermosalient compounds.

One of the least supported points in the interpretation of the results appears to be the calculation of the shear strain. I was confused by the paragraph "The anisotropic lattice expansion in the (010) plane generates a shear force parallel to the unit cell diagonals, resulting in a huge macroscopic shear deformation parallel to the crystal sides of more than 18% (Fig. 2f and Supplementary Fig. S4). This large strain is comparable with biological muscles (~20%) and two times larger than that of thermally activated shape memory alloys (~8.5%)". First, the expansion in a plane (2D) alone can not account for the shear along the unit cell (3D) diagonals, the formulation of the first part of this paragraph is not very precise, and it should be reformulated. Moreover, looking at the change in the (010) plane (using the non-standard setting in Table 1), from the main text alone, I was not sure how was the shear strain calculated. I found the explanation in the caption of the Supplementary Figure S4, however, I am not convinced that this way to calculate the shear strain and the result is directly comparable to the values calculated or measured for muscles and shape memory alloys. I would suggest that the calculations of the shear strain be removed completely from the main text and the supplementary information to avoid running into wrong conclusions by comparing values that are not directly comparable to each other.

The sentence "crystal splitting parallel to the (010) plane was observed in a few crystals which may be because of the weak intermolecular force between the layers along the normal direction; upon cooling, their shape still completely recovered, but not the damage due to the splitting". What was the ratio of the crystals that split compared to those that didn't? Include that number in the main text, which will provide further information on the robustness of this material.

Figure 3b: I am surprised that the authors decided to use crystal corner angles to demonstrate the cyclability. Angles can be measured, but it is more common (and also possible with greater precision) to measure distances. Please replace this plot with a plot that shows the length of one or two crystal sides or another length (the plot showing the change of the angle can be included in the supplementary information), which will also show better whether there is a fatigue over time. Include standard deviations.

Section "Temperature controlled movement of load": The authors have used a very clever way to demonstrate the performance of the actuator. How does this compare with the recently characterized molecular crystal actuators reported by Khalil et al. (oi.org/10.1021/jacs.8b12752) and Li et al. (10.1039/C9SC02444A)? Add comment and comparison in this section of the main text.

Stylistic improvements:

I believe that the words "microactuator", "thermosalient" and "thermoelastic" should be written without a hyphen.

Reference 9 is not properly cited (check the author names).

Reference 14: Although I mentioned this work as part of my first review as one of the many examples, this reference is not relevant to this work and should be removed (the same with the

text in lines 264 and 265).

Figure 2, panels a – d: it is not clear where panels a and b and panels c and d start. This part of the figure can be rearranged by leaving some space between the panels a, b, c and d.

The word “perfectly” used in the abstract may be too strong within the context of a scientific claim, and should be removed.

Line 100: The last name of this person is Kobatake, and Seiya is his first name. Please correct.

Table 1 should be moved to the supplementary material.

Table 1: the e.s.d.s for the structure at 170 oC should be included in the table.

Another relevant article with a report of a molecular crystal that can be used as actuator was published in the meantime by Li and the collaborators, and it should be included in the introduction and in the discussion: Chem. Sci., DOI: 10.1039/C9SC02444A

The sentence starting with “We believe” in the abstract should be rephrased. The readers are not interested in what is concluded based on the data presented, rather than in what the authors believe.

Line 60: the verb “can form” should be replaced with “crystallizes as” or a similar phrase.

Figure 3: the symbols “alpha” and “beta” on the crystals are huge and their size should be reduced.

Line 68: “in a kind of” is rather vague expression to express the fact of whether the structure is layered (or not), and it should be removed.

Reference 24: The last names of the authors are not properly cited.

Caption of Figure 1: “for the rotation of the flat planes in the molecules” should be rephrased, because the planes can not rotate, only parts of a molecule or molecules can do.

Line 151: “parallelogram crystal sides” does not sound very crystallographic, and should be reformulated

Figure 3a: the label on the y axis should be “Heat flow” instead of “Heating flow”

Reviewer #2 (Remarks to the Author):

At first, this can be published on Nature Communications with respect to the correctness of the scientific description by responding to almost all inquiries by reviewers including me besides the novelty of the paper got rather common by the deeper comprehension of the material's behavior during revision.

However, I appreciate the authors' truthful attitude to science.

Reviewer #3 (Remarks to the Author):

I carefully read the revised manuscript; the authors have addressed all the raised concerns. I can recommend this manuscript to be accepted to Nature Communications

Reviewer #4 (Remarks to the Author):

Most of my previous critical comments were taken into account by authors in revised manuscript which I found suitable for publication in Nature Communications.

Reviewer #1 (Remarks to the Author):

In the first revised version of the manuscript as well as in their response to the reviewers' comments, the authors have provided stronger claims in support of the uniqueness of their material, and I am now convinced that this work warrants publication in Nature Communications. The combination of a thermosalient effect (which needs to be discussed in more detail), the robustness upon phase transition, the elegant molecular dynamics simulations, and the demonstration of application of this material together are important and interesting, and I believe that after another round of revision this manuscript will be acceptable for publication. In the revised version, the main claims were strengthened significantly by including the crystal structures of the two phases, and the mechanism of the phase transition is now much clearer. The reference list is now also more inclusive, and except for a couple of very recent publications (see my specific comments provided below), it reflects better the state of the art in this research. The authors may consider the following comments to improve their manuscript further. I also noticed that in their response to the first set of comments, the authors did respond however they did not make all of the changes in the manuscript; the authors are strongly encouraged to respond to the comments by making actual changes in the manuscript for each and every comment in addition to addressing the reviewers' comments in their response letter.

Reply 1: We are glad that the reviewer found our modifications convincing to consider our manuscript to be suitable for publication in Nature Communications. We think that our manuscript may be further improved after revision according to some of the reviewer's comments. Concerning the requested references, we noticed that one was published the day before we got the reviewer's comments (Li et al., 24 July 2019). We think it is unreasonable to include this reference. Concerning the other reference (Khalil et al. Feb 2019), we already included it as Suppl. Ref 3 in the resubmitted manuscript on comparison of strain. Since this study does not report the crystallographic information of the transition (we have another reference of the same material that includes the crystallographic information, Ref. 3), it is hard to make a direct comparison with the material.

Most of the main text is dedicated to explanation of the mechanism of the phase transition, and that is definitely one of the most important aspects of this work. However, not much is said about the thermosalient effect per se. A reader who is not familiar with this topic might question whether this compound is really thermosalient material or not. The discussion is reduced to only one sentence stating one crystal jumps and a figure in the supplementary information. How do the crystals move? How many of them move and how many don't? With what speed and in which direction? How many retain their integrity and how many of them disintegrate? Please add a short discussion and include more details on the thermosalient effect at the beginning of the discussion section.

Reply 2: As mentioned in the text, crystals only jumped when placed on their side faces. In many cases, they jumped out of the field of view of the microscope, which makes it very hard to follow. Moreover, the goal of our paper is to investigate the mechanism underlying the thermosalient effect, since several papers (refs. 1-5) already looked at the mechanical/jumping aspect. Since the phase transition mechanism is independent of the orientation of the crystal, we studied most crystals while lying on their basal plane when they do not jump. This makes it much easier to follow the transition mechanism.

Introduction section, the first sentence: this sentence should be modified, because not all thermosalient transitions proceed in a single-crystal-to-single crystal manner.

Reply 3: Indeed we were too specific. The corresponding sentence has been modified (Line 28-29)

The introduction, the sentence starting with "The transition appears to be triggered by small conformational changes in the molecules...". This goes well with some previous observations, and references with previous observations where the same conclusions were obtained should be added here.

Reply 4: Thanks for pointing this out and we have added the references (refs. 2, 3).

Two other molecular crystals that were characterized as single crystal actuators were published recently (DOI: 10.1039/C9SC02444A, DOI: org/10.1021/jacs.8b12752). These articles are directly related to this work, and they should be included in the introduction and in the discussion. A comparison between the two materials and the one studied here should be added in the discussion of the main text, in the section "Shear deformation and durability".

Reply 5: Please see reply 1.

In other thermosalient materials, usually the volume of the high-temperature phase is very slightly bigger (up to several percent) compared to that of the low-temperature one. In case of the material studied here, however, it looks like it is smaller. The authors should provide a brief explanation of this result and the reasons for the reduction of the volume.

Reply 6: Indeed, in our material the cell volume of the high-temperature phase (HT) is slightly smaller than that of the low-temperature phase (LT). We believe this is because the molecular configuration of the HT is more planar than that of the LT, which allows for a denser molecular packing. We added a brief explanation in Line 177-179.

This article reports a thermosalient effect, however the only figure that shows the effect is in the Supplementary material. The readers would benefit from a figure which shows the effect in the main text. This can be added to the current Figure 1.

Reply 7: Please, see reply 2 for our motivation.

Panel 1b: The contours, which represent a projection of the crystal shape, might be confused with a unit cell. A contour of the actual unit cell should be added on the plots of both phases to distinguish it from the crystal shape.

Reply 8: Thank you. This is a good point, and we have modified Fig.1b accordingly.

Discussion section: The discovery of a new thermosalient compound which does not appear to belong to any of the three classes that were established previously by Sahoo et al., is a particularly novel and important aspect of this work, and this point of novelty is definitely something that brings this work within the scope of Nature Communications. The readers will benefit from a short discussion that explains the difference in the intermolecular interactions and molecular shape of this compound with other thermosalient compounds.

Reply 9: We have added this discussion in the main text (Line 113-118). There is no strong interaction between the molecules within the layers and this allows the phase boundary to be perpendicular to the layer plane, in contrast with two of the classes by Sahoo *et al.* The third

class includes transitions with hindered rotation of bulky groups. Although torsional rotation plays a role in the present transition, it is not hindered since the dihedrals do not change sign.

One of the least supported points in the interpretation of the results appears to be the calculation of the shear strain. I was confused by the paragraph “The anisotropic lattice expansion in the (010) plane generates a shear force parallel to the unit cell diagonals, resulting in a huge macroscopic shear deformation parallel to the crystal sides of more than 18% (Fig. 2f and Supplementary Fig. S4). This large strain is comparable with biological muscles (~20%) and two times larger than that of thermally activated shape memory alloys (~8.5%)”. First, the expansion in a plane (2D) alone can not account for the shear along the unit cell (3D) diagonals, the formulation of the first part of this paragraph is not very precise, and it should be reformulated. Moreover, looking at the change in the (010) plane (using the non-standard setting in Table 1), from the main text alone, I was not sure how was the shear strain calculated. I found the explanation in the caption of the Supplementary Figure S4, however, I am not convinced that this way to calculate the shear strain and the result is directly comparable to the values calculated or measured for muscles and shape memory alloys. I would suggest that the calculations of the shear strain be removed completely from the main text and the supplementary information to avoid running into wrong conclusions by comparing values that are not directly comparable to each other.

Reply 10: We agree that a direct comparison of the shear deformation of our crystal with that in biological muscles is not very suitable, so we deleted the corresponding sentence. However, we can compare the strain with shape memory alloys, as they have a similar phase transition mechanism (martensitic phase transition), which also proceeds via a shear deformation.

To clarify the calculation of the strain we added references to Suppl. Fig. S4 in the main text.

The sentence “crystal splitting parallel to the (010) plane was observed in a few crystals which may be because of the weak intermolecular force between the layers along the normal direction; upon cooling, their shape still completely recovered, but not the damage due to the splitting”. What was the ratio of the crystals that split compared to those that didn’t? Include that number in the main text, which will provide further information on the robustness of this material.

Reply 11: The crystal splitting case was indeed observed, yet very rare. We have only seen it in two crystals out of the 50+ studied. We added this to the text, Line 180.

Figure 3b: I am surprised that the authors decided to use crystal corner angles to demonstrate the cyclability. Angles can be measured, but it is more common (and also possible with greater precision) to measure distances. Please replace this plot with a plot that shows the length of one or two crystal sides or another length (the plot showing the change of the angle can be included in the supplementary information), which will also show better whether there is a fatigue over time. Include standard deviations.

Reply 12: Measuring the corner angle change is the most correct and accurate way in our case, as we study shear shape change and we can compare phase transitions of different crystals (e.g. Fig.1 and Fig. S2, the corner angles of every crystal are the same).

Section “Temperature controlled movement of load”: The authors have used a very clever way to demonstrate the performance of the actuator. How does this compare with the recently characterized molecular crystal actuators reported by Khalil et al. ([oi.org/10.1021/jacs.8b12752](https://doi.org/10.1021/jacs.8b12752)) and Li et al. (10.1039/C9SC02444A)? Add comment and comparison in this section of the main

text.

Reply 13: Please, see reply 1.

Stylistic improvements:

Reply 14: We thank the reviewer for his/her time and efforts. We have followed most of stylistic improvements and the changes are highlighted in the main text. We did not follow the following three comments:

Table 1 should be moved to the supplementary material.

Reply 16: Moving Table 1 to the SI also involves adding most of the content to the main text which does not aid the readability of the manuscript.

Table 1: the e.s.d.s for the structure at 170 oC should be included in the table.

Reply 17: As mentioned in the text, we have only performed a unit cell determination at 170 °C, with limited reflections. The accuracy of the cell lengths is reflected by the limited significant digits provided.

REVIEWERS' COMMENTS:

Reviewer #1 (Remarks to the Author):

The authors have made an effort to address this reviewer's comments, and in many cases, that has been done successfully. There are remaining points that were not addressed and would normally need to be clarified, however I feel that these changes will not affect much the essence of the research results presented in the manuscript. Given the need for urgent publication of the important results presented in this manuscript, I can now recommend it for publication without further changes.